# Text-to-Unlearn: Robust Concept Removal in GANs via Text Prompts

## Abstract

State-of-the-art generative models exhibit powerful image-generation capabilities, raising ethical and legal challenges for service providers. Consequently, Content Removal Techniques (CRTs) have emerged to control outputs without requiring full retraining. However, the problem of unlearning in Generative Adversarial Networks (GANs) remains largely unexplored. We propose Text-to-Unlearn, a novel framework that selectively unlearns concepts from pre-trained GANs using only text prompts, enabling feature and identity unlearning, as well as fine-grained tasks such as expression and multi-attribute removal in models trained on human faces. Our approach leverages natural language descriptions to guide unlearning without additional datasets or supervised finetuning, offering a scalable solution. To evaluate the effectiveness of our method, we introduce an automated unlearning assessment method using state-of-the-art image-text alignment metrics and propose a new metric: *degree of unlearning*. Additionally, we assess robustness by introducing a prompt boundary attack to subvert unlearning. Our results demonstrate that Text-to-Unlearn achieves robust unlearning, resisting adversarial attempts to recover erased concepts while preserving model utility. To our knowledge, this is the first cross-modal unlearning framework for GANs, advancing the management of generative model behavior.

## 1 Introduction

Generative image models have revolutionized image synthesis, enabling applications ranging from AI art generation to digital humans for Virtual Reality (VR) technology with unprecedented realism. These models have significantly empowered independent content creators by lowering the barrier to producing high-quality artwork. Furthermore, several popular services like ChatGPT and Grok now support integrated instructional image editing, bringing advanced image synthesis to the fingertips of anyone with a smartphone. However, the widespread adoption of these models has also made it easier for malicious actors to generate deepfakes containing harmful imagery and sensitive content, potentially violating regulations such as the GDPR and CCPA (Mantelero, 2013; Goldman, 2020). To address this issue, Content Removal Techniques (CRTs) have emerged as a way to selectively remove learned concepts from models, ensuring fine-grained control over model outputs without damaging overall performance. CRTs for generative image models can be broadly categorized into *filtering* and *unlearning*-based strategies. Filtering strategies rely on trained classifiers or heuristic rules to detect and block undesirable outputs, without altering the underlying model parameters. They are lightweight and suitable in dynamic settings (*e.g.*, identifying NSFW content from a prompt). Unlearning-based strategies involve finetuning the model to address the root cause. They are particularly relevant to compliance issues when service providers cannot rely on filtering strategies that may fail to identify undesirable content.

Deep Neural Network (DNN)-based image generation relies on two primary model architectures: Generative Adversarial Networks (GANs) and diffusion models (Goodfellow et al., 2014; Podell et al., 2024; Ho et al., 2020). Text-to-image diffusion models, with their high-quality image synthesis, have spurred extensive research into security and privacy challenges, including Content Removal Techniques (CRTs). While recent work on concept removal has therefore focused on diffusion models, GANs remain widely used in production. They power various image editing and interactive applications such as Adobe Photoshop's neural filters library, AR/VR environments, and avatar systems, enabling fine-grained editing (*e.g.*, adjusting makeup, age, or expressions). Their advan-

tages include: (i) image generation in a single forward pass, offering faster inference than diffusion models, (ii) resource efficiency, and (iii) precise control via latent space manipulation (Wu et al., 2021; Kocasari et al., 2022; Bermano et al., 2022). If a GAN model trained on faces or identities is challenged under GDPR/CCPA, providers must remove specific individuals or features without retraining. This is the research problem we address in this paper.

Existing work on unlearning in GANs has largely been limited to single-attribute erasure and often depends on carefully curated datasets, limiting scalability (Seo et al., 2024; Moon et al., 2024). Unlike diffusion models with built-in text conditioning, GANs lack native textual control, making targeted concept removal more complex, yet necessary, as GANs are embedded in several interactive applications. To address this critical research gap, we introduce Text-to-Unlearn: a cross-modal framework for robust concept removal in GANs. Our contributions are listed below:

- A novel framework that removes learned concepts in GANs using *only* a text prompt, eliminating the need for additional datasets.

- A new quantitative evaluation metric, *degree of unlearning* ($\gamma$), designed to measure unlearning using state-of-the-art Vision-Language Models (VLMs).

- A systematic method to quantitatively evaluate the robustness of unlearning using a novel prompt boundary attack.

## 2 RELATED WORK

Recent research has addressed the potential misuse of generative image models such as StyleGAN2, Stable Diffusion, and DALL-E2 through generative unlearning (Karras et al., 2020; Podell et al., 2024; Ramesh et al., 2022). Seo et al. (2024) present GUIDE, which focuses on identity unlearning in GANs using a reference image, enabling removal of unseen identities via a latent target method and an adjacency-aware loss. This approach ensures neighboring latent points map to different identities post-unlearning while maintaining the GAN's overall utility. However, our work tackles a broader challenge, using text prompts to unlearn diverse features not necessarily adjacent in the latent space. Additionally, Moon et al. (2024) explore feature unlearning in VAEs and GANs, relying on curated datasets (*e.g.*, CelebA) and frameworks like Morpho-MNIST for annotations to finetune models. (de Castro et al., 2019) In contrast, we address unlearning with fewer assumptions, applying it to large, high-resolution datasets like FFHQ where data curation is impractical due to privacy constraints.

Beyond GAN-specific efforts, unlearning and concept erasure have been studied in diffusion models. Safe Latent Diffusion (SLD) mitigates harmful content at inference without finetuning, while recent methods like MACE, Erased Stable Diffusion (ESD), and Forget-Me-Not propose finetuning approaches for concept erasure. Our method broadly falls into the latter category (Schramowski et al., 2023; Lu et al., 2024; Gandikota et al., 2023; Zhang et al., 2024).

## 3 PROBLEM STATEMENT

Unlike existing unlearning research, class labels may not always be available or relevant for generative models (Moon et al., 2024; Tiwary et al., 2025). Additionally, collecting images to make datasets for unlearning can be challenging due to privacy regulations like GDPR and CCPA. Thus, we are motivated by the following question: *Can we flexibly unlearn concepts from a GAN using only text prompts?*

As shown in StyleCLIP, text prompts can be used to drive image manipulation in the $\mathcal{W}^+$ latent space of the GAN to make various edits by leveraging CLIP's joint embedding space, and thus, overcoming the hurdle of finding the right directions in the $\mathcal{W}^+$ latent space (Patashnik et al., 2021; Radford et al., 2021). As such, the ability to unlearn must be just as flexible as the image generation process. Unlearning for GANs involves finetuning the model to prevent the generation of targeted concepts, such as specific features or identities while preserving overall image quality. Formally, we consider a pre-trained StyleGAN generator $G_t(\theta_0)$, where $\theta_0$ represents the initial model parameters. Given a text prompt $p$ describing the target concept, we seek an unlearning strategy $\Lambda$ that yields a

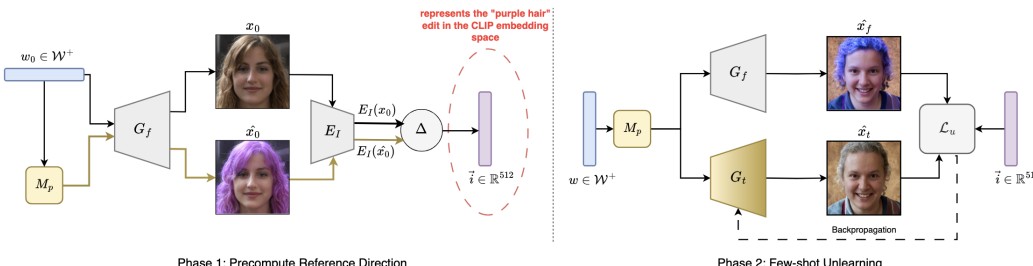

Figure 1: Overview of the Text-to-Unlearn framework for unlearning the feature "purple hair" as an example. In the first phase, a reference direction to guide the unlearning is precomputed once. In the second phase, the precomputed reference direction is used to steer the trainable generator's synthesis network away from generating undesirable images.

modified generator $G_t(\theta)$ :

$$G_t(\theta) \triangleq \Lambda(G_t(\theta_0), p) \tag{1}$$

The updated generator $G_t(\theta)$ must exclude the target concept from its outputs while maintaining high-quality generation for other concepts.

## 3.1 THREAT MODEL

**Attack Scenario.** We consider an adversary with white-box access to the unlearned StyleGAN generator and the StyleGAN generator before unlearning. The adversary's goal is to circumvent any unlearning strategies and regenerate images containing unlearned concepts from the unlearned model. Following related work, the adversary does not need access to the original dataset in our threat model (Hu et al., 2024). In terms of capabilities, the adversary can craft adversarial text prompts that guide the image generation process to force unlearned concepts to appear in the output images.

**Scope.** Our threat model focuses on StyleGAN's $\mathcal{W}^+$ latent space and our proposed Text-to-Unlearn scheme. We exclude attacks requiring access to training data, as these are less feasible in typical deployment scenarios, and black-box attacks, which are less effective given the adversary's white-box access.

## 3.2 CHALLENGES

Given the scope of our problem and threat model, unlearning in GANs presents unique challenges to secure generative modeling:

- Erasing entire concepts instead of a single image from the GAN's latent space without affecting the overall image synthesis quality is difficult due to entanglement in the latent space, *i.e.*, erasing one concept can easily affect the generation of several other features.

- Unlike diffusion models, GANs do not have textual inputs to generate samples for the unlearning process. Finding interpretable directions in a pre-trained GAN's latent space for each dataset is intractable at scale.

- A key challenge specific to unlearning for GANs is the difficulty in measuring the extent to which unlearning is successful because it can be subjective.

- Besides the tradeoff between degree of unlearning and model utility, we need to account for a third dimension, namely, resilience to the adversarial attacks discussed above.

## 4 METHODOLOGY

In this section, we first discuss the components of our framework (shown in Figure 1) and then introduce one of our core contributions: *directional unlearning*. Our methodology is motivated by the few-shot domain adaptation scheme in StyleGAN-NADA (Gal et al., 2022).

### 4.1 OVERVIEW

Our framework consists of four key components: a latent mapper $M_p$ trained on a text prompt $p$ that describes the concept to be unlearned, a frozen copy of the generator $G_f$, a trainable generator $G_t$, and a pre-trained CLIP model.

**Latent Mapper ($M_p$).** The latent mapper $M_p$ (as described in StyleCLIP) is a shallow neural network that maps latent codes within StyleGAN's $\mathcal{W}^+$ space, *i.e.*, $M_p : \mathcal{W}^+ \to \mathcal{W}^+$ and is used to edit images according to the prompt $p$. Suppose a point $w \in \mathcal{W}^+$ corresponds to an image of a man with black hair and the text prompt $p$ is "purple hair". Then, the latent mapper can be used to compute $\hat{w} = w + M_p(w)$ such that $\hat{w}$ corresponds to an image of the same person with the only difference being purple hair. Simply put, we can use the latent mapper to edit any image according to a text description.

**Generators.** The trainable GAN generator $G_t$ will be finetuned using our unlearning strategy and will no longer produce images with the unlearned feature after the unlearning process is complete. $G_f$ is a copy of $G_t$ before unlearning and is used to generate images containing the feature to be unlearned.

**Pre-trained CLIP model ($E$).** We use a pre-trained CLIP model $E$, leveraging its visual encoder $E_I$ to compute image embeddings.

### 4.2 DIRECTIONAL UNLEARNING

The overall idea is to finetune $G_t$ using a few samples taken from its latent space so that it does not produce images containing the undesirable properties described by the text prompt. Since GANs are prone to mode collapse, $G_f$ and $M_p$ are used to help generate specific images, which are used to regularize the training with appropriate loss components. After the finetuning (unlearning) is complete, $G_f$ can be discarded. Our unlearning process is based on guiding $G_t$ along a direction in the CLIP embedding space derived from the text prompt $p$, so we deem our method as *directional unlearning*. The process is split into two phases: (i) Phase 1: Precomputing a reference direction for unlearning, and (ii) Phase 2: Few-shot Unlearning.

**Phase 1.** We choose a randomly sampled batch of latent codes $w_0 \in \mathcal{W}^+$ called the initial latent codes. The latent mapper uses them to compute $\hat{w}_0 = w_0 + M_p(w_0)$. The corresponding image batches for $w_0$ and $\hat{w}_0$ are then given by $x_0$ and $\hat{x}_0$ (*e.g.*, images with purple hair):

$$x_0 = G_f(w_0), \; \hat{x}_0 = G_f(\hat{w}_0) \tag{2}$$

Once the pairs of image batches are computed, we compute a unit vector $\vec{i}$ (operation denoted as $\Delta$ in Figure 1) representing the edit direction in the CLIP embedding space. Specifically,

$$\vec{i} = \frac{E_I(\hat{x}_0) - E_I(x_0)}{\|E_I(\hat{x}_0) - E_I(x_0)\|_2} \tag{3}$$

where $E_I(\cdot)$ represents the CLIP visual encoder, and $E_I(x_0)$ and $E_I(\hat{x}_0)$ are the CLIP embeddings of the image batches $x_0$ and $\hat{x}_0$, respectively. Essentially, we capture the change of adding $p$ (*e.g.*,"purple hair") in the embedding space and later unlearn along this direction.

**Phase 2.** Now, we perform few-shot unlearning using the reference direction $\vec{i}$ (from Equation 3) from Phase 1. During each finetuning step, a batch of latent codes $w \in \mathcal{W}^+$ is sampled and passed through the latent mapper $M_p$ to generate latent codes $\hat{w} = w + M_p(w)$. The latent codes $\hat{w}$ will be provided as input to $G_t$ and $G_f$. We freeze the generator (as done in StyleGAN-NADA) to ensure the optimization remains on the real image manifold (Gal et al., 2022). During the finetuning process, $G_f$ will constantly generate images with the target concept described by $p$. However, $G_t$ will adapt to generate the same images without the target concept because the loss function $\mathcal{L}_u$ uses the precomputed reference direction $\vec{i}$ to guide the unlearning only along this direction.

**Loss Function.** First, we define the unlearning loss function $\mathcal{L}_u$ for *feature unlearning*:

$$\begin{aligned}\mathcal{L}_u = \mathcal{L}_{dir}(\hat{x}_t, \hat{x}_f, \vec{i}) + \lambda_{lpips}\mathcal{L}_{lpips}(\hat{x}_t, \hat{x}_f) + \\ \lambda_{id}\mathcal{L}_{id}(\hat{x}_t, \hat{x}_f)\end{aligned} \tag{4}$$

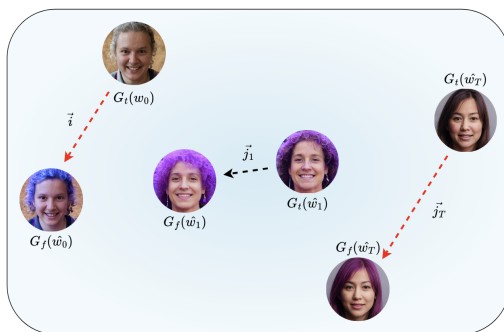

Figure 2: Examples of image embeddings in the CLIP space during the fine-tuning of $G_t$. $\vec{i}$ is the precomputed reference direction and, $\vec{j_1}$ and $\vec{j_T}$ are alignments during and at the end of training, respectively.

Here, $\mathcal{L}_{dir}$ is the directional loss, $\mathcal{L}_{lpips}$ is the LPIPS loss for perceptual similarity, and $\mathcal{L}_{id}$ is an ID loss based on the ArcFace facial recognition network (Zhang et al., 2018; Deng et al., 2022). $\hat{x_f}$ and $\hat{x_t}$ are the images generated by $G_f$ and $G_t$, respectively. $\vec{i}$ is the precomputed reference direction from Equation 3. The directional loss $\mathcal{L}_{dir}$ is the key component that guides the trainable generator $G_t$ away from synthesizing the target concept. However, while unlearning the features, we need to preserve the usability of the latent space for downstream tasks, and thus, we regularize the training process using ID loss and LPIPS loss.

Suppose that $d_{cos}(\cdot)$ represents the cosine similarity function, then the directional loss is defined as:

$$\mathcal{L}_{dir}(\hat{x}_t, \hat{x}_f, \vec{i}) = 1 - d_{cos}(\vec{i}, \vec{j})$$
$$\vec{j} = \frac{E_I(\hat{x}_t) - E_I(\hat{x}_f)}{\|E_I(\hat{x}_t) - E_I(\hat{x}_f)\|_2} \tag{5}$$

The unit vector $\vec{j}$ captures the difference between outputs of $G_t$ and $G_f$ in the CLIP embedding space. During unlearning, we align $\vec{j}$ with the fixed precomputed reference direction $\vec{i}$. Minimizing $\mathcal{L}_{dir}$ rewards this alignment, progressively excluding the undesired feature from $Gt$ outputs. To illustrate, consider the example in Figure 2. For initial latents $w_0$, $\vec{i}$ is computed per Equation 3. At the first training batch $\hat{w}_1$, $G_t(\hat{w}_1)$ retains less prominent purple hair, so $\vec{j_1}$ misaligns with $\vec{i}$. By final step $T$, $G_t(\hat{w}_T)$ excludes purple hair entirely, yielding perfect alignment of $j_T$ with $\vec{i}$. Here, the inputs to $G_f$ are fixed latents for purple-hair images, and alignment occurs only if $G_t$ learns *along* $\vec{i}$ to synthesize purple-hair-free images.

In the case of *identity unlearning*, we formulate a different unlearning loss $\mathcal{L}_{u,id}$ such that $G_t$ directs images toward the mean latent. Here, we only use the LPIPS loss to ensure optimization favors images from the original domain. Suppose the mean latent is given by $\overline{w} \in \mathcal{W}^+$ and the corresponding image is $\overline{x} = G_t(\overline{w})$, then the unlearning loss is given by:

$$\mathcal{L}_{u,id} = \mathcal{L}_{dir}(\hat{x}_t, \overline{x}, \vec{i}_{id}) + \mathcal{L}_{lpips}(\hat{x}_t, \overline{x}) \tag{6}$$

We define $\vec{i}_{id}$ as the precomputed reference direction for identity unlearning, which is computed with respect to the mean latent as shown in Equation 7.

$$\vec{i}_{id} = \frac{E_I(x_0) - E_I(\overline{x})}{\|E_I(x_0) - E_I(\overline{x})\|_2}, \quad \vec{j}_{id} = \frac{E_I(\hat{x}_t) - E_I(\overline{x})}{\|E_I(\hat{x}_t) - E_I(\overline{x})\|_2} \tag{7}$$

Here, $x_0$ is the batch of images randomly sampled at the start of Phase 1. The underlying optimization problem remains the same as before.

## 5 SUBVERTING UNLEARNING: PROMPT BOUNDARY ATTACK

Text-to-Unlearn leverages CLIP's joint embedding space for unlearning, so the adversary may try and use the same space to find prompts that are semantically similar or "close" to the original prompt.

We propose an optimization-based approximation strategy that operates in the continuous embedding space, perturbing token embeddings and iteratively optimizing them to approximate the target boundary while regularizing to ensure the resulting embeddings can be projected to valid tokens in the discrete space. Let $E_t : \mathcal{T} \to \mathbb{R}^d$ denote CLIP's text encoder, which maps discrete token sequences $\mathbf{s} \in \mathcal{T}$ from the set of valid token sequences $\mathcal{T}$ to continuous embeddings in CLIP's $d$ dimensional embedding space $\mathbb{R}^d$. For an original prompt's token sequence $\mathbf{s}_0$, it's CLIP embedding is $\mathbf{e}_0 = E_t(\mathbf{s}_0)$. We seek a perturbed embedding $\mathbf{e}_p$ such that $\|\mathbf{e}_p - \mathbf{e}_0\|_\infty = \epsilon$, where $\epsilon$ is the perturbation strength and $\mathbf{e}_p$ corresponds to a valid token sequence $\mathbf{s}_p \in \mathcal{T}$. Since optimization over discrete tokens is non-differentiable, we optimize continuous token embeddings $\mathbf{t} = [\mathbf{t}_1, \ldots, \mathbf{t}_m] \in \mathbb{R}^{m \times h}$ where $m$ is the number of non-padded tokens and $h$ is the embedding dimension. For the sake of simplicity, we present the constrained optimization problem for a single perturbation in Equation 8:

$$\min_{t \in \mathbb{R}^{m \times h}} \mathcal{L}(\mathbf{t}) = \alpha \left\| \tilde{E}_t(\mathbf{t}) - \mathbf{e}_p \right\|_2^2 + \beta \sum_{i=1}^m \left\| \mathbf{t}_i - \mathbf{t}_i^0 \right\|_2^2$$

$$\text{s.t.} \left\| \tilde{E}_t(\mathbf{t}) - \mathbf{e}_0 \right\|_\infty = \epsilon \tag{8}$$

Here, $\mathbf{t}_i^0$ is the continuous embedding of the $i$-th token in $\mathbf{s}_0$, derived from $E_t$'s token embedding matrix $\mathbf{W} \in \mathbb{R}^{|\mathcal{V}| \times h}$. $\mathcal{V}$ is the vocabulary set of of $E_t$. $\tilde{E}_t(\cdot)$ is a part of the same text encoder $E_t$ that maps continuous token embeddings to $R^d$. $\mathbf{e}_p$ is obtained by dense sampling on the $\ell_\infty$-norm ball surface according to Equation 9:

$$\mathbf{e}_p = \mathbf{e}_0 + \epsilon \cdot \mathbf{u}, \quad \text{where } \|\mathbf{u}\|_\infty = 1 \tag{9}$$

$\alpha$ and $\beta$ are coefficients to scale the importance of the loss components. However, our goal is to find $\mathbf{s}_p$, so we project the optimal continuous token embedding, say $\mathbf{t}^*$, obtained from Equation 8 using Equation 10, where $\mathbf{W}_{\mathbf{s}_i}$ refers to the vector represented by the $\mathbf{s}_i$-th row of the token embedding matrix $\mathbf{W}$:

$$\mathbf{s}_p = \arg\min_{s \in \mathcal{T}} \sum_{i=1}^m \|\mathbf{t}_i^* - \mathbf{W}_{\mathbf{s}_i}\|_2^2 \tag{10}$$

Additionally, the latent boundary attack, a more straightforward yet general strategy to inspect the $\mathcal{W}^+$ space, is detailed in Appendix B, offering further insights into the robustness of the method.

## 6 EXPERIMENTS

### 6.1 EXPERIMENTAL SETUP

We consider two types of evaluation scenarios: *in-domain* and *out-of-domain* experiments. In-domain experiments evaluate the effectiveness of unlearning on samples generated from the latent space of the GAN. Out-of-domain experiments evaluate how well the unlearning strategy generalizes by testing on samples from the CelebA-HQ dataset. The images from the CelebA-HQ dataset are encoded into the $\mathcal{W}^+$ space using the e4e encoder, and we evaluate the effectiveness of unlearning on these samples (Tov et al., 2021). In each evaluation scenario, we use 1000 images for testing. Tested concepts include visual features (*e.g.*, "purple hair", "spectacles", "mohawk hairstyle"), identities (*e.g.*, "Taylor Swift"), and some nonstandard unlearning tasks. All training details are available in the Appendix.

### 6.2 QUALITATIVE RESULTS FOR UNLEARNING

**Feature Unlearning.** We consider unlearning the following features of varying granularity: hair color, hairstyle, and accessories. The results of the GAN before and after unlearning are shown in Figure 3, and we see that for any chosen source image, the latent mapper can generate an edit with the target feature. Using our text-guided unlearning scheme, the latent codes of images with undesirable features are now mapped to variations of the source image without those features.

**Identity Unlearning.** Our Text-to-Unlearn framework relies solely on text prompts, limiting its scope to unlearning identities accessible via CLIP's text encoder (presumably seen during pre-training). The results of unlearning identities using Equation 6 are shown in Figure 3. In contrast

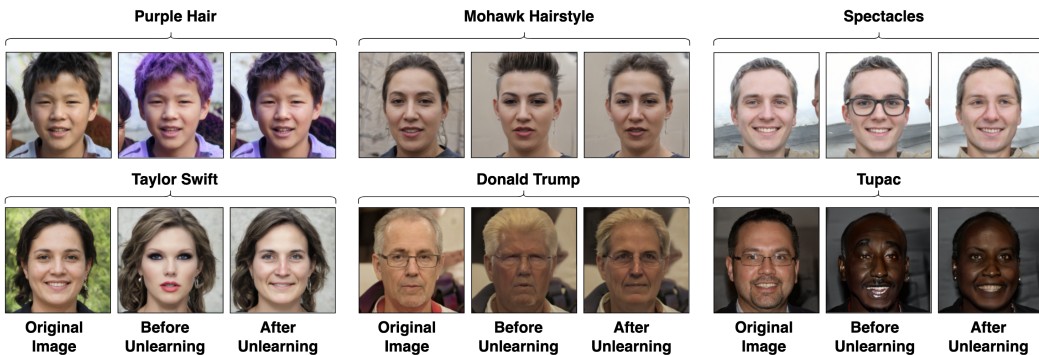

Figure 3: Comparison of images before and after unlearning concepts based on text prompts.

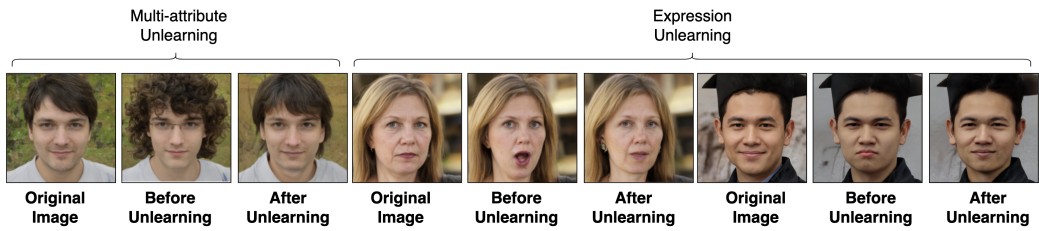

Figure 4: Examples of non-standard unlearning tasks including multi-attribute and expression unlearning. First trio: "curly long hair", second trio: "surprised", and third trio: "angry".

to feature unlearning, identity unlearning results in images that diverge more noticeably from the original source images because we direct them toward the mean latent during training.

**Nonstandard Unlearning Tasks.** We leverage the disentangled $\mathcal{W}^+$ space to perform expression unlearning and multi-attribute unlearning. The key advantage of Text-to-Unlearn is the flexibility of text prompts, which allow unlearning multiple undesired features via a single prompt. Similarly, we can also unlearn expressions from the model. The results for the unlearning prompts "curly long hair", "surprised", and "angry" are shown in Figure 4. After unlearning the features, we inspect the usability of the GAN for downstream tasks like StyleCLIP image manipulation. We present some example manipulations using the latent mapper in Figure 5 after unlearning "purple hair" (left) and "spectacles" (right). We see that the GAN cannot generate purple hair even after using a new latent mapper trained on the prompt "purple hair". However, other edits can be made without training new latent mappers.

## 6.3 EVALUATION METRICS

We want to quantitatively evaluate unlearning in GANs using our Text-to-Unlearn framework, but existing metrics like FID and IS evaluate image fidelity and are not suitable for evaluating unlearning (Heusel et al., 2017; Salimans et al., 2016). Instead, we formulate this problem as measuring

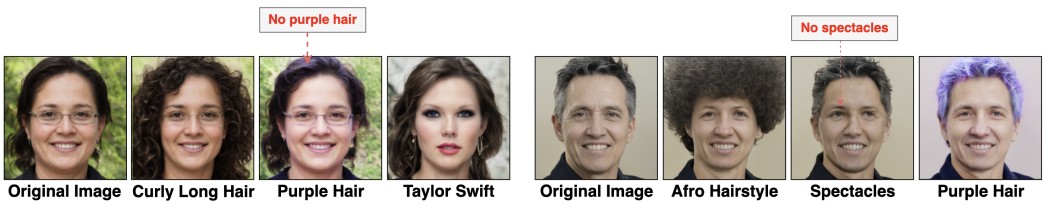

Figure 5: Example of using latent mappers to make edits after unlearning purple hair (left) and spectacles (right) from the GAN. The manipulation prompt is listed below each image.

Table 1: Degree of unlearning ($\gamma$) computed using various image-text alignment scoring metrics for *in-domain* and *out-of-domain* images. Higher scores are better ($\uparrow$) and are highlighted in bold.

| Text Prompt | CLIP-FlanT5 ($\uparrow$) | | | | LLaVA-1.5 ($\uparrow$) | | | | BLIP-2 ($\uparrow$) | | | |
|---|---|---|---|---|---|---|---|---|---|---|---|---|
| | In-Domain | | Out-of-Domain | | In-Domain | | Out-of-Domain | | In-Domain | | Out-of-Domain | |
| | Baseline | Ours | Baseline | Ours | Baseline | Ours | Baseline | Ours | Baseline | Ours | Baseline | Ours |
| Purple Hair | 0.26 | **0.74** | 0.38 | **0.88** | 0.46 | **0.88** | 0.60 | **0.80** | 0.39 | **0.77** | 0.76 | **0.83** |
| Mohawk Hairstyle | 0.37 | **0.81** | 0.67 | **0.94** | 0.84 | **0.88** | 0.87 | **0.94** | 0.65 | **0.65** | 0.78 | **0.78** |
| Spectacles | 0.03 | **0.73** | 0.43 | **0.55** | 0.02 | **0.87** | 0.01 | **0.64** | 0.16 | **0.84** | 0.21 | **0.29** |
| Curly Long Hair | 0.36 | **0.85** | 0.56 | **0.98** | 0.32 | **0.73** | 0.43 | **0.88** | 0.48 | **0.99** | 0.70 | **0.98** |
| Surprised | 0.50 | **0.76** | 0.66 | **0.72** | 0.31 | **0.70** | 0.46 | **0.73** | 0.42 | **0.78** | 0.62 | **0.95** |
| Angry | 0.10 | **0.62** | 0.20 | **0.82** | 0.16 | **0.84** | 0.25 | **0.92** | 0.17 | **0.81** | 0.25 | **0.96** |
| Afro Hairstyle | 0.62 | **0.89** | 0.65 | **0.82** | 0.59 | **0.96** | 0.68 | **0.89** | 0.51 | **0.99** | 0.74 | **0.94** |
| Makeup | 0.14 | **0.89** | 0.26 | **0.99** | 0.12 | **0.86** | 0.21 | **0.97** | 0.18 | **0.51** | 0.60 | **0.62** |
| Bobcut Hairstyle | 0.69 | **0.70** | 0.59 | **0.80** | 0.35 | **0.39** | 0.35 | **0.56** | 0.36 | **0.40** | 0.57 | **0.66** |

Table 2: ID scores for the baseline method and our method after unlearning different identities computed using 5000 samples. Lower scores are better ($\downarrow$).

| Prompt | Taylor Swift | Donald Trump | Tupac Shakur |
|---|---|---|---|
| ID (Baseline) $\downarrow$ | 0.38 | 0.82 | 0.88 |
| ID (Ours) $\downarrow$ | **0.2** | **0.3** | **0.5** |

the alignment of the unlearning prompt $p$ with images from the trainable generator $G_t$ before and after unlearning. Recent work has extensively explored the problem of measuring image-text alignment and moving beyond simple alignment metrics like CLIP score. (Lu et al., 2023; Lin et al., 2024; Singh & Zheng, 2023) We use the BLIP-2 ITM Score, VQAScore, and IDScore to evaluate unlearning. Detailed descriptions of the metrics are available in Appendix A.2.

## 6.4 QUANTITATIVE RESULTS FOR UNLEARNING

**Baseline.** Since there is no relevant work that uses only text to unlearn from GANs, we employ an intuitive baseline method: We use a latent mapper to generate images with the target concept from $G_t$, and maximize the CLIP loss with respect to the unlearning prompt. The loss function minimized during unlearning is given in Equation 11:

$$\mathcal{L}_{baseline} = -\mathcal{L}_{CLIP}(\hat{x}_t, p) \tag{11}$$

Here, $\mathcal{L}_{CLIP}(\cdot, \cdot)$ is the CLIP loss between an image and text prompt, $\hat{x}_t$ is the synthesized image during training, and $p$ is the prompt. This approach does not use the directional loss from our method.

**Evaluation Method.** For each text prompt, we use a latent mapper to sample 1000 in-domain images from the GAN before and after unlearning. Initially, most contain the target concept, but post-unlearning, few do. We compute score distributions using CLIP-FlanT5 VQAScore, LLaVA VQAScore, and BLIP-2 ITM metrics for these sets, plus a reference distribution from 1000 unrelated images (as unrelated image-text pairs often yield non-zero scores). Our goal is to maximize separation between pre-unlearning and post-unlearning distributions, yielding the degree of unlearning metric $\gamma$ in Equation 12:

$$\gamma = \frac{W_1(A, B)}{W_1(B, R)}, \tag{12}$$

where $W_1(\cdot, \cdot)$ is the Wasserstein-1 distance between distributions $A$ (after unlearning), $B$ (before unlearning), and $R$ (reference). We use the 1-Wasserstein distance because it is suitable for ordered data and does not rely on assumptions about the score distributions. Additional details are available in Appendix C.1.

For out-of-domain assessment, we encode 1000 CelebAHQ images into the latent space using the e4e encoder and compute analogous distributions, verifying the generalization of unlearning to out-of-domain data (Table 1). Clearly, directional unlearning outperforms the baseline across all prompts. The average ID scores for identities (Table 2) highlight that the similarity to the target identity is lower after unlearning using our method.

Table 3: Evaluation of unlearning robustness ($\epsilon = 0.15$). Each row shows an example adversarial prompt and mean CLIP scores for images generated using the original prompt, adversarial prompts before and after unlearning, and unrelated prompts. Adversarial prompts are meaningful, with scores close to the original prompt before unlearning. Low scores after unlearning and high robustness indicate effective unlearning.

| Original Prompt | Example Adversarial Prompt | Original Prompt Score Before Unlearning | Original Prompt Score After Unlearning | Adversarial Score Before Unlearning | Adversarial Score After Unlearning | Unrelated Prompt Scores | $R_p$ |
|---|---|---|---|---|---|---|---|
| Taylor Swift | "primaries swift" | $0.283 \pm 0.037$ | $0.130 \pm 0.007$ | $0.283 \pm 0.009$ | $0.148 \pm 0.004$ | $0.140 \pm 0.029$ | 0.947 |
| Donald Trump | "grapptrump" | $0.311 \pm 0.028$ | $0.188 \pm 0.010$ | $0.312 \pm 0.003$ | $0.187 \pm 0.003$ | $0.185 \pm 0.014$ | 0.981 |
| Purple Hair | "purpletail" | $0.281 \pm 0.022$ | $0.163 \pm 0.013$ | $0.263 \pm 0.008$ | $0.193 \pm 0.017$ | $0.176 \pm 0.019$ | 0.804 |
| Mohawk Hairstyle | "usable mohawk tious" | $0.238 \pm 0.034$ | $0.19 \pm 0.017$ | $0.268 \pm 0.025$ | $0.212 \pm 0.012$ | $0.191 \pm 0.017$ | 0.727 |
| Spectacles | "codenspectacles" | $0.233 \pm 0.027$ | $0.173 \pm 0.013$ | $0.243 \pm 0.008$ | $0.185 \pm 0.009$ | $0.179 \pm 0.014$ | 0.917 |

## 6.5 ROBUSTNESS EVALUATION

To assess robustness, we evaluate resilience against the prompt boundary attack from Section 5. We select five test prompts (Table 3) and, for each, generate five perturbed variants (e.g., "primaries swift" for "Taylor Swift") within an $\ell_\infty$-norm ball of radius $\epsilon = 0.15$. We train latent mappers on these adversarial prompts and validate them by ensuring they produce the same edits as the original prompts pre-unlearning, which is necessary for meaningful evaluation. Robustness is then measured by the extent to which these prompts recover the erased concept. Additional qualitative results are available in Appendix B.1.2 along with a detailed perturbation budget analysis in Appendix B.3.

We generate 250 images for each of four groups: images from edits using a latent mapper trained on the original prompt before unlearning, images from edits using a latent mapper trained on the adversarial prompt before unlearning, images from edits using a latent mapper trained on the adversarial prompt after unlearning, and a control group paired with unrelated text prompts. The control group addresses non-zero scores from image-text alignment metrics for unrelated pairs. We calculate a robustness score, $R_p$, against the prompt boundary attack using Equation 13:

$$R_p = 1 - \frac{\mu_u - \mu_r}{\mu_o - \mu_r} \in [0, 1] \tag{13}$$

Here, $\mu_u$ represents the mean CLIP score of images after unlearning, $\mu_o$ is the mean CLIP score of the original images before unlearning, and $\mu_r$ is the mean CLIP score of the control group, computed using unrelated prompts. Based on Equation 13, we see that the robustness is evaluated based on how the mean CLIP score changes with respect to the control as a result of unlearning. A higher $R_p$ score is indicative of robust unlearning.

As shown in Table 3, the Text-to-Unlearn framework is resilient to the prompt boundary attack across various concepts, achieving a high $R_p$ score. The "mohawk hairstyle" has a slightly lower $R_p$ score compared to the other prompts. We inspect the images generated using the adversarial mapper after unlearning and find that the concept is erased. We conjecture that the high CLIP score after unlearning is due to CLIP's text encoder struggling to capture the specificity of a true mohawk hairstyle from other hairstyles. A detailed analysis is included in Appendix B.1.1.

## 7 LIMITATIONS, CONCLUSION, AND FUTURE WORK

In this paper, we introduce Text-to-Unlearn, a framework to unlearn concepts from a GAN using only text prompts. Our experiments show that Text-to-Unlearn can achieve favorable results at different levels of granularity, which we validate using the degree of unlearning ($\gamma$). Notably, we demonstrate the effectiveness of our approach on a high-resolution GAN, moving beyond typical analyses on datasets like CIFAR-10 and MNIST to emphasize its practical applicability. Furthermore, we show the resilience of our unlearning strategy against a novel prompt boundary attack and the latent boundary attack. We do acknowledge that the framework's reliance on CLIP may limit the unlearning of poorly represented concepts or introduce biases. In our future work, we would like to explore how to build unlearning strategies that work with debiased VLMs (Hirota et al., 2024; Berg et al., 2022).

## 8 REPRODUCIBILITY STATEMENT

We do not propose new datasets. The unlearning only relies on samples from the GAN's latent space, so there is no need for any external datasets/splits. All hyperparameters and hardware requirements for experiments are provided in the appendix. The algorithm for the prompt boundary attack is also provided.

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

# A    ADDITIONAL DETAILS ABOUT EXPERIMENTS

## A.1    MODEL CONFIGURATION

We use a StyleGAN2 model pre-trained on the FFHQ dataset at $1024\times1024$ resolution and a CLIP model with the ViT-B/32 configuration for text-guided unlearning and evaluation. All experiments are conducted on a single NVIDIA A100 GPU. No external datasets were used, with all samples for unlearning being drawn directly from the GAN's latent space, aligning with our goal of using only text for unlearning.

## A.2    METRICS

In Section 6, we mentioned three metrics used to evaluate unlearning. We provide more details below:

**BLIP-2 ITM Score.** BLIP-2 is a multimodal model similar to CLIP that has a joint embedding space for images and text. We use the Image-Text Matching (ITM) score to evaluate our method.

**VQAScore.** VQA models are designed to answer questions about images. We evaluate the image-text alignment by querying the model with the question "Does this figure show {*text*}? Please answer yes or no." The VQAScore is computed as the probability that the answer is yes given a question and image, i.e., P("Yes" | question, image). Despite being simplistic, it has been shown to outperform several image-text alignment baselines and achieve state-of-the-art results. We use CLIP-FlanT5 XL and LLaVA-1.5 7B Liu et al. (2024) to generate the VQAScores.

**ID Score.** For identity unlearning, we utilize a latent mapper to identify latent codes of images that exhibit features of the target identity. Following the unlearning process, we analyze the modified images using the ArcFace neural network Deng et al. (2022). Specifically, we calculate the cosine similarity between the features extracted from the images before and after unlearning to evaluate how effectively the identity has been removed.

## A.3    LATENT MAPPER TRAINING

Here, we provide detailed instructions and hyperparameters used for training the latent mapper. Unlike StyleCLIP, we train the latent mapper on samples from the latent space of the GAN since we do not use external datasets. Our hyperparameters are different from StyleCLIP for certain prompts.

There are 3 hyperparameters for the latent mapper training: (i) ID loss regularization ($\lambda_{ID}$), (ii) L2 loss regularization ($\lambda_{L2}$), and (iii) Step magnitude in the $\mathcal{W}^+$ space ($\delta$). In practice, the latent mapper is implemented as $\hat{w} = w + \delta \cdot M_p(w)$ to ensure gradients are updated stably. The training parameters are listed in Table 4.

| Text Prompt | $\lambda_{ID}$ | $\lambda_{L2}$ | $\delta$ | Levels |
|---|---|---|---|---|
| Purple Hair | 0.1 | 0.8 | 0.1 | fine, medium, coarse |
| Mohawk Hairstyle | 0.1 | 0.8 | 0.8 | medium, coarse |
| Spectacles | 0.1 | 0.8 | 0.9 | medium, coarse |
| Curly Long Hair | 0.1 | 0.8 | 0.8 | medium, coarse |
| Surprised | 0.1 | 0.8 | 0.5 | medium, coarse, fine |
| Angry | 0.1 | 0.8 | 0.3 | medium, coarse, fine |
| Afro Hairstyle | 0.1 | 0.8 | 0.8 | medium, coarse |
| Makeup | 0.1 | 0.8 | 0.3 | medium, coarse, fine |
| Bobcut Hairstyle | 0.1 | 0.8 | 0.3 | medium, coarse |
| Taylor Swift | 0 | 0.8 | 0.1 | fine, medium, coarse |
| Donald Trump | 0 | 1.5 | 0.1 | fine, medium, coarse |
| Tupac Shakur | 0 | 1.5 | 0.1 | fine, medium, coarse |

Table 4: Hyperparameters for training the latent mapper.

| Text Prompt | $lr$ | $\lambda_{ID}$ | $\lambda_{lpips}$ |
|---|---|---|---|
| Purple Hair | 8e-3 | 4e-1 | 1e-1 |
| Mohawk Hairstyle | 8e-3 | 4e-1 | 1e-1 |
| Spectacles | 1e-2 | 2e-1 | 1e-1 |
| Curly Long Hair | 8e-3 | 4e-1 | 1e-1 |
| Surprised | 8e-3 | 4e-1 | 1e-1 |
| Angry | 8e-3 | 4e-1 | 1e-1 |
| Afro Hairstyle | 8e-3 | 4e-1 | 1e-1 |
| Makeup | 8e-3 | 4e-1 | 1e-1 |
| Bobcut Hairstyle | 8e-3 | 4e-1 | 1e-1 |
| Taylor Swift | 8e-3 | 0 | 1e-1 |
| Donald Trump | 8e-3 | 0 | 1e-1 |
| Tupac Shakur | 8e-3 | 0 | 1e-1 |

Table 5: Hyperparameters for unlearning.

## A.4 UNLEARNING HYPERPARAMETERS

For the directional unlearning procedure, we have three hyperparameters: (i) Learning Rate ($lr$), (ii) ID loss regularization ($\lambda_{ID}$), and (iii) LPIPS loss regularization ($\lambda_{lpips}$). The hyperparameters to reproduce our results are listed in Table 5. $\lambda_{ID}$ is 0 for the identity prompts since this is not a loss component for identity unlearning as discussed in the main paper.

## B ADDITIONAL DETAILS ABOUT SUBVERSION ATTACKS

### B.1 PROMPT BOUNDARY ATTACK

In this section, we provide some intuition about the prompt boundary attack to supplement the material in the main paper. The basic idea of this attack is to exploit the reliance on CLIP for unlearning and craft adversarial prompts to generate images with unlearned concepts. For example, if the prompt "Taylor Swift" was used to train the latent mapper for unlearning, the adversary could use similar prompts like "T4yl0r Sw1ft" to train a latent mapper and find latent codes of images that resemble Taylor Swift. This attack is based on the observation that text captions and images in the CLIP space have a many-to-many relationship, *i.e.*, many text phrases are related to the same image and vice-versa.

Consider a naive strategy: The adversary can manually craft perturbations of the target concept to be recovered. As an example, we consider perturbations of the prompt "Taylor Swift" made using leet-speak: "T4yl0r Sw1ft" and "Tay1or 5wift". However, these prompts are only valid candidates for the attack if they can be used to generate images similar to the original prompt (*i.e.*, "Taylor Swift"). In Figure 6, we show the results of training latent mappers on the aforementioned leet-speak prompts. Clearly, the edits do not resemble the mapper trained on the original prompt "Taylor

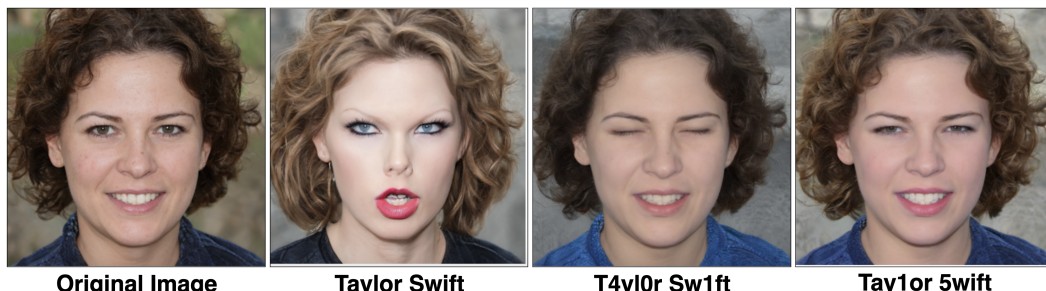

| **Original Image** | **Taylor Swift** | **T4yl0r Sw1ft** | **Tay1or 5wift** |

Figure 6: Examples of edits made using latent mappers trained on naive adversarial prompts compared to the original prompt "Taylor Swift". The text prompt used to train the latent mapper is listed below each image.

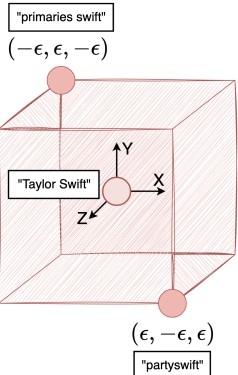

Figure 7: Examples of edits made using latent mappers trained on naive adversarial prompts compared to the original prompt "Taylor Swift". The text prompt used to train the latent mapper is listed below each image.

Swift". The leet-speak prompts might be similar to the original prompts to readers, but do not share sufficient semantic similarity in the CLIP embedding space, and thus, they do not produce the same images. So, we ask ourselves *Can we find adversarial prompts to subvert unlearning in a scalable manner?*

As mentioned in the Section 5, the adversary may try to find semantically similar prompts. We illustrate this concept in Figure 7, which depicts adversarial prompts lying on the boundary of an $\ell_\infty$-norm ball centered around the original prompt "Taylor Swift". Note that the $\ell_\infty$-norm ball is shaped like a cube in Figure 7 because the $\ell_\infty$-norm constrains perturbations by the largest absolute coordinate, uniformly bounding each dimension to yield a hypercubic geometry. The adversary's goal is to find prompts corresponding to points on this norm ball's surface, as these represent the maximum allowable perturbation while remaining semantically close. However, CLIP's text encoder $E_t$ is a one-way function that maps discrete text prompts, which are represented as integer token IDs, to continuous real-valued embeddings. This is the problem we highlight in the main paper.

The full algorithm is provided in Algorithm 1.

We also provide a full list of adversarial prompts used for the robustness evaluation in Table 6.

In Figure 8, we present some results using the prompt boundary to show that it is indeed possible to subvert unlearning.

### B.1.1 DISCUSSION ABOUT ROBUSTNESS FOR "MOHAWK HAISRTYLE"

In this section, we inspect some of the images used to evaluate the $R_p$ score for the prompt "mohawk hairstyle". After unlearning, the latent mapper trained using the adversarial prompt "usable mohawk

---

**Algorithm 1** Prompt Boundary Attack: Finding Perturbed Prompts on the Prompt Boundary

---

**Require:** Original prompt $\mathbf{s}_0 \in \mathcal{T}$, CLIP text encoder $E_t$, token embedding matrix $\mathbf{W} \in \mathbb{R}^{|\mathcal{V}| \times h}$, perturbation strength $\epsilon$, number of perturbations $N$, iterations $n_{\text{iters}}$, learning rate $\eta$, hyperparameters $\alpha, \beta$

1: **Output**: Perturbed token sequences $\{\mathbf{s}_p^{(j)}\}_{j=1}^N$
2: $\mathbf{e}_0 \leftarrow E_t(\mathbf{s}_0)$                        ▷ Compute original embedding
3: $\mathbf{t}^0 \leftarrow \mathbf{W}[\mathbf{s}_0[1:m],:]$          ▷ Original token embeddings for $m$ non-padded tokens
4: **for** $j = 1$ to $N$ **do**                     ▷ Dense sampling on $\ell_\infty$-norm ball
5:      $\mathbf{e}_p^{(j)} \leftarrow \mathbf{e}_0 + \epsilon \cdot \mathbf{u}, \quad \|\mathbf{u}\|_\infty = 1$
6: **end for**
7: Initialize $\mathbf{T} = [\mathbf{t}^{(1)}, \ldots, \mathbf{t}^{(N)}] \in \mathbb{R}^{N \times m \times h}$ randomly
8: **for** $i = 1$ to $n_{\text{iters}}$ **do**
9:      $\mathbf{T} \leftarrow \mathbf{T} - \eta \nabla_\mathbf{T} \mathcal{L}(\mathbf{T})$    ▷ Gradient descent update
       where $\mathcal{L}(\mathbf{T}) =$

$$\frac{1}{N} \sum_{j=1}^N \left( \alpha \|\tilde{E}_t(\mathbf{t}^{(j)}) - \mathbf{e}_p^{(j)}\|_2^2 + \beta \|\mathbf{t}^{(j)} - \mathbf{t}^0\|_F^2 \right)$$

10: **end for**
11: $\mathbf{S}_p \leftarrow []$                                         ▷ Store results
12: **for** $j = 1$ to $N$ **do**
13:      $\mathbf{s}_p^{(j)} \leftarrow \arg\min_{\mathbf{s} \in \mathcal{T}} \sum_{i=1}^m \|\mathbf{t}_i^{(j)} - \mathbf{W}_{\mathbf{s}_i}\|_2^2$
14:      Append $\mathbf{s}_p^{(j)}$ to $\mathbf{S}_p$
15: **end for**
16: **return** $\mathbf{S}_p$

---

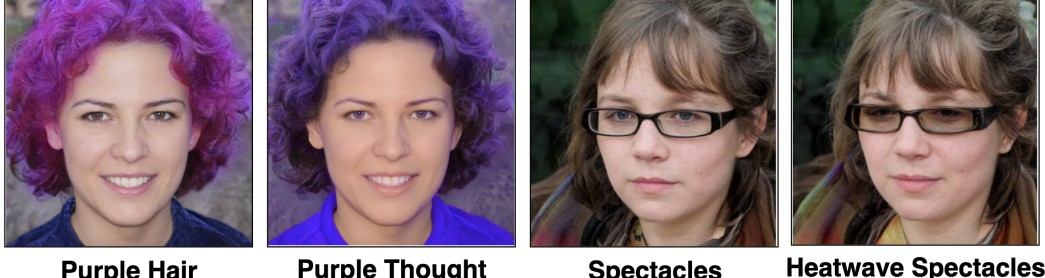

**Purple Hair**      **Purple Thought**      **Spectacles**      **Heatwave Spectacles**

Figure 8: Edits made using latent mappers trained on adversarial prompts for "purple hair" and "spectacles".

Table 6: Adversarial prompts for each original prompt used in the prompt boundary attack.

| Original Prompt | Adversarial Prompts |
|---|---|
| Taylor Swift | primaries swift, taylor ♀, partyswift, steamer taylor swift, needle taylor swift |
| Donald Trump | grapptrump, advancetrump, donald trump mastered, eagle donald trump, gray trump |
| purple hair | purpletail, purple thought, petals hair, purple aring, purple pai |
| mohawk hairstyle | usable mohawk tious, mohawk braided, mohawk predictions, babs mohawk skater, mohawk appli |
| spectacles | codenspectacles, heatwave spectacles, bl spectacles, spectacles moron, down spectacles yt |

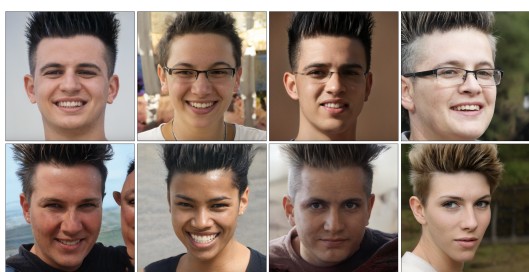

Figure 9: Images generated using the prompt "usable mohawk tious" before unlearning.

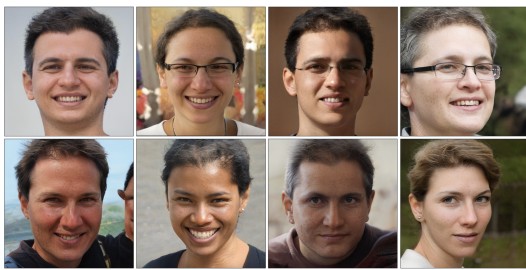

Figure 10: Images generated using the prompt "usable mohawk tious" after unlearning.

tious" generates images with the highest average CLIP score. We compare the images generated using this adversarial prompt before and after unlearning in Figures 9 and 10. Clearly, the images in Figure 10 resemble the mohawk images in Figure 3, which have a high degree of unlearning (shown in Table 1). The text prompt "mohawk hairstyle" is well-represented in the CLIP embedding space and understood by the text encoder, and thus, the latent mappers (regular and adversarial) are able to generate the mohawk hairstyle before unlearning. However, after unlearning, the CLIP model likely associates some artifacts or loosely related features to the text prompt "mohawk hairstyle", which results in a higher mean CLIP score after unlearning despite successful concept erasure.

### B.1.2 ADDITIONAL RESULTS FOR PROMPT BOUNDARY ATTACK

In this section, we present additional qualitative results showcasing the effectiveness of latent mappers trained on adversarial prompts on the GAN before unlearning. Essentially, we want to validate that these adversarial prompts on the "boundary" of the original prompt are indeed valid threats by virtue of being able to produce the same edits as the original prompt. We consider adversarial prompts of "purple hair", "Taylor Swift", and "spectacles" and present the results in Figure 11.

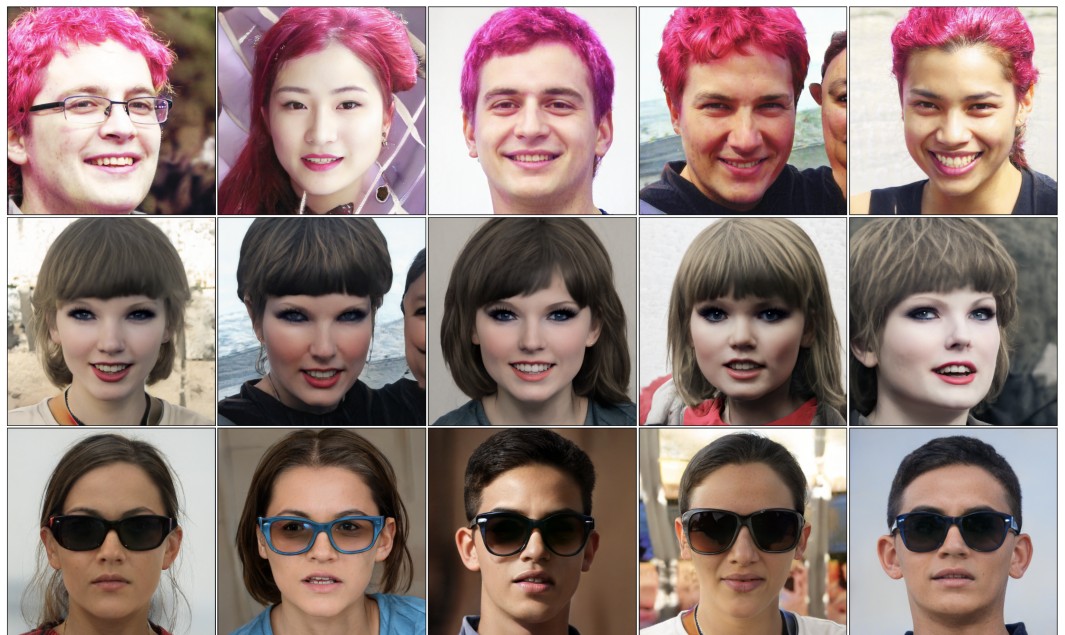

Figure 11: Images generated using mappers trained on adversarial prompts: (i) "petals hair" (top row), (ii) "primaries swift" (middle row), and (iii) "codenspectacles" (bottom row).

## B.2 LATENT BOUNDARY ATTACK

The latent boundary attack aims to subvert the unlearning process in a GAN by recovering unlearned concepts through targeted perturbations in the $\mathcal{W}^+$ latent space. This attack exploits potential residual traces of the unlearned concept that may persist in the generator's latent space after unlearning. The adversary is assumed to have a latent mapper trained on the target concept. The attack operates by exploring the $\mathcal{W}^+$ latent space around initial latent codes that, prior to unlearning, would produce images with the target concept. To find these initial latent codes, the adversary randomly generates several latent codes, which may or may not have the target concept. The latent mapper is applied to these latent codes to derive the desired initial latent codes of images containing the target concept. To probe for vulnerabilities, the adversary conducts a search by densely sampling points on the $\ell_\infty$-norm ball around each of the initial latents. If the attack is successful, the latent codes on the norm ball will produce images with the unlearned concept. This is a relatively straightforward attack compared to the prompt boundary attack.

For the latent boundary attack, we select five test prompts, as listed in Table 7. For each prompt, we train a latent mapper to generate the corresponding edits in the $\mathcal{W}^+$ space. Using the latent mapper, we sample 100 latent codes (images), and for each image, we sample 100 perturbations on the surface of the $\ell_\infty$-norm ball around the latent before and after unlearning. We evaluate the robustness against the latent boundary attack by measuring the unlearning for perturbed latents relative to a baseline of images without the unlearned concept, similar to the prompt boundary attack. The robustness score, $R_l$, is computed using Equation 14:

$$R_l = 1 - \frac{\mu'_u - \mu_r}{\mu'_o - \mu_r} \in [0, 1] \tag{14}$$

Here, $\mu'_o$ is the mean CLIP score of perturbed latents before unlearning, $\mu'_u$ is the mean CLIP score of perturbed latents after unlearning, and $\mu_r$ is the mean CLIP score of the control group.

## B.3 PERTURBATION BUDGET ANALYSIS

The robustness evaluation is only meaningful if an appropriate perturbation strength $\epsilon$ is chosen. Consider the latent boundary attack: a very small perturbation might yield high robustness scores but may not account for valid adversarial latents obtained using a higher perturbation strength. On

Table 7: Evaluation of unlearning robustness using latent perturbations ($\epsilon = 0.3$). Each row corresponds to a prompt, showing CLIP scores for the original latent, and mean CLIP scores for perturbed latents before and after unlearning. Scores for perturbed latents are close to the original latent before unlearning. Lower scores after unlearning and a high robustness metric indicate effective unlearning.

| Original Prompt | Original Latent Score Before Unlearning | Original Latent Score After Unlearning | Perturbed Latent Score Before Unlearning | Perturbed Latent Score After Unlearning | Unrelated Prompt Scores | $R_l$ |
|---|---|---|---|---|---|---|
| Taylor Swift | $0.287 \pm 0.0271$ | $0.142 \pm 0.0141$ | $0.276 \pm 0.0283$ | $0.144 \pm 0.0166$ | $0.141 \pm 0.0256$ | 0.980 |
| Donald Trump | $0.306 \pm 0.0257$ | $0.186 \pm 0.0106$ | $0.303 \pm 0.0256$ | $0.191 \pm 0.0134$ | $0.187 \pm 0.0156$ | 0.971 |
| Purple Hair | $0.277 \pm 0.0285$ | $0.178 \pm 0.0153$ | $0.272 \pm 0.0287$ | $0.182 \pm 0.0167$ | $0.176 \pm 0.0175$ | 0.937 |
| Mohawk Hairstyle | $0.251 \pm 0.0266$ | $0.198 \pm 0.0155$ | $0.249 \pm 0.0272$ | $0.199 \pm 0.0166$ | $0.193 \pm 0.0195$ | 0.881 |
| Spectacles | $0.244 \pm 0.0229$ | $0.170 \pm 0.0124$ | $0.242 \pm 0.0250$ | $0.180 \pm 0.0138$ | $0.179 \pm 0.0139$ | 0.985 |

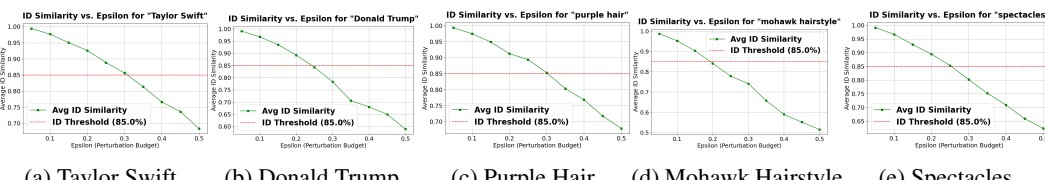

(a) Taylor Swift     (b) Donald Trump     (c) Purple Hair     (d) Mohawk Hairstyle     (e) Spectacles

Figure 12: Perturbation budget ($\epsilon$) analysis for the latent boundary attack using ID Score (Threshold $\tau = 0.85$) for select test prompts. The prompt is listed below each graph.

the other hand, an excessively high perturbation budget might yield a low robustness score even if the unlearning is successful, as the perturbed latents may not represent realistic variations. Therefore, for the latent boundary attack, where perturbations are made in a continuous latent space, a perturbation budget analysis is essential to determine the upper bound on $\epsilon$ such that the evaluation is valid and representative of the worst-case scenario while inspecting latents around the $\ell_\infty$-norm ball for the target concept. We use the ID similarity score to set this upper bound, selecting $\epsilon$ values where the similarity score remains above a threshold of $\tau = 0.85$, ensuring the perturbed latents retain the target concept across various identities. Larger perturbations may produce latents corresponding to different identities or images outside the real image manifold; we mitigate the former by sampling multiple latent codes and disregard the latter as irrelevant. In contrast, for the prompt boundary attack, the discrete nature of text prompts and the non-continuous relationship between perturbation strength and adversarial impact make a traditional perturbation budget analysis less applicable. However, we find that for $\epsilon > 0.15$, the latent mappers trained on these adversarial prompts do not generate edits similar to the original prompt, much like Figure 6. In Figure 12, we present the results of the perturbation budget analysis for the latent boundary attack.

## C    ADDITIONAL DETAILS AND EXPERIMENTS FOR UNLEARNING

### C.1    VQASCORE DISTRIBUTIONS AND VISUALIZING DoU

In this section, we provide some intuition about how the degree of unlearning metric ($\gamma$) captures the "shift" in score distributions. In Figures 13 and 14, the red histogram represents the VQAScore distribution of images after unlearning ($A$ from Section 6), the blue histogram represents the VQAScore distribution of images before unlearning ($B$ from Section 6), and the green histogram represents the VQAScore distribution of random images unrelated to the target feature ($R$). Upon successful unlearning, the blue and red histograms should be as separated as possible. The red histogram should be similar to the green distribution, resembling images without the target concept. As seen in Figures 13 and 14, our method achieves superior separation compared to the baseline method.

### C.2    EVALUATING DISENTANGLEMENT OF UNLEARNING

In this section, we evaluate how disentangled our unlearning method is, *i.e.*, we evaluate the extent to which our unlearning method affects the image generation of other features. First, we sample 400 images per feature for a set of four features ("purple hair", "spectacles", "surprised", "afro

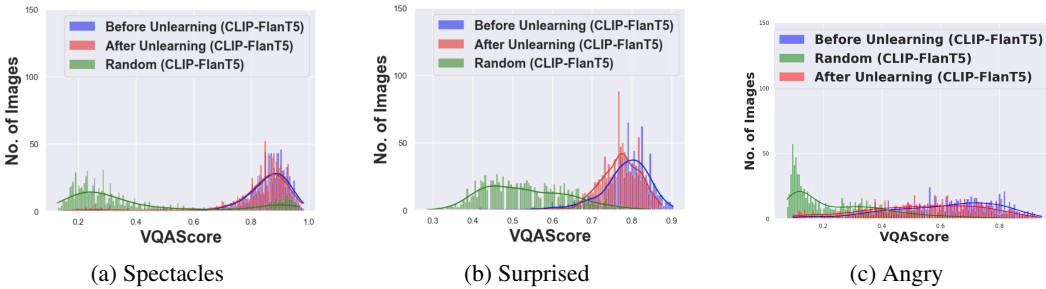

(a) Spectacles                (b) Surprised                (c) Angry

Figure 13: CLIP-FlanT5 VQAScore distribution computed over 1000 images before and after unlearning for different text prompts using the baseline method.

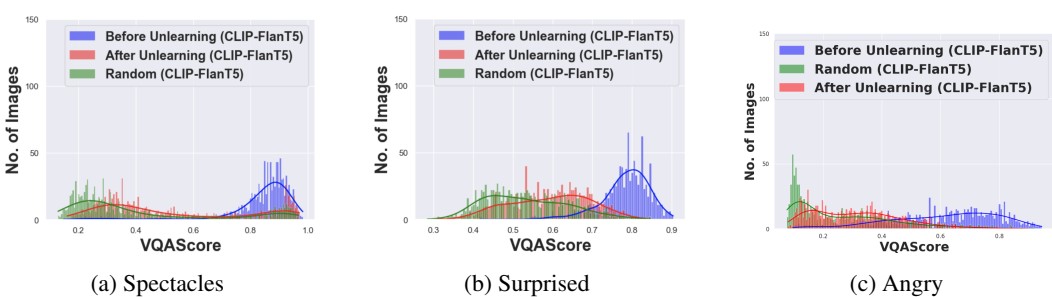

(a) Spectacles                (b) Surprised                (c) Angry

Figure 14: CLIP-FlanT5 VQAScore distribution computed over 1000 images before and after unlearning for different text prompts using our *directional unlearning* method.

hairstyle") from the GAN prior to unlearning and compute the average VQAScore for each prompt as a baseline. Then, we unlearn each feature and evaluate the change in the mean VQAScore for the other three features. In Table 8, we report the shift in mean VQAScores from the baseline. We see that there is a marginal shift in the scores for unrelated features, suggesting that our method supports disentangled unlearning.

## C.3    ABLATION STUDY

We perform three ablation experiments (as shown in Figure 15): impact of (i) loss function components, (ii) batch size of the automatic layer selection strategy in Phase 2, and (iii) batch size used when computing the reference direction $\vec{i}$ (in Phase 1) on the degree of unlearning. The ideal batch size, for the automatic layer selection strategy and for computing the reference direction $\vec{i}$ is 8 based on the stability across all prompts. We also see that both the LPIPS loss and ID loss help achieve maximal unlearning.

## C.4    EVALUATING MODEL UTILITY

In Table 9, we evaluate the fidelity of image generation after unlearning. We see that our Text-to-Unlearn framework has better FID scores compared to the baseline. A higher FID score does not

Table 8: Quantitative results for the effect of unlearning each feature (rows) on the VQAScore of other unrelated features (columns). Each entry is a percentage change of the CLIP-FlanT5 VQAScore for that feature with respect to its baseline score before unlearning.

| Feature | Purple Hair | Spectacles | Surprised | Afro Hairstyle |
|---|---|---|---|---|
| Purple Hair | -60% | +1.2% | -0.2% | -0.2% |
| Spectacles | -0.4% | -30.4% | +1% | -1% |
| Surprised | -0.4% | +1% | -20% | -1.1% |
| Afro Hairstyle | -0.7% | +0.7% | +0.8% | -44.8% |

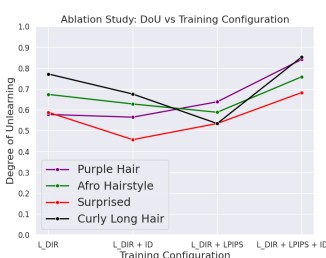 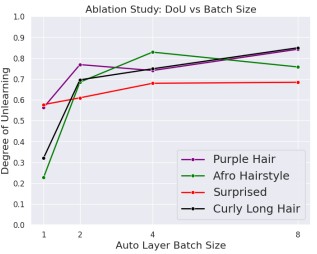 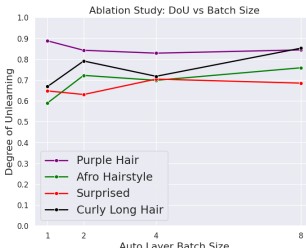

(a) Ablation on loss components (directional loss, ID loss, and LPIPS loss) of $\mathcal{L}_u$.

(b) Ablation on batch size used for computing the reference direction $\vec{i}$ in Phase 1.

(c) Ablation on batch size used for automatic layer selection in Phase 2.

Figure 15: Ablation experiments for relevant hyperparameters.

Table 9: Impact of unlearning strategies on the FID score of the GAN for select prompts. The FID scores are computed using 50,000 samples with the FFHQ dataset as a reference. Lower scores are better ($\downarrow$).

| Prompt | FID ($\downarrow$) | |
|---|---|---|
| | Ours | Baseline |
| Purple Hair | **7.84** | 28.74 |
| Mohawk Hairstyle | **5.87** | 32.55 |
| Spectacles | **8.97** | 32.21 |
| Taylor Swift | **22.64** | 40.67 |
| Donald Trump | **17.19** | 41.93 |

necessarily mean the quality of image generation is worse. In general, unlearning a concept increases the FID score because it penalizes the lack of diversity compared to the original FFHQ dataset. In the case of identity unlearning, the FID scores are higher because we direct the target latents toward the mean latent face, reducing the diversity of images compared to feature unlearning. However, the large difference in FID scores after unlearning the same concept using different unlearning strategies suggests a difference in image quality. We refer readers to Appendix C.5 for some examples.

C.5 DISCUSSION ABOUT TRAINING STABILITY

Here, we discuss the stability provided by *directional unlearning* during the unlearning process. Based on Figure 13, one could think of increasing the learning rate for the baseline method to achieve better unlearning. Figure 16 shows the results of unlearning after 400 and 700 steps. Using our *directional unlearning* method, we can subtly unlearn the "angry" expression whereas the baseline method causes distortion in the images generated. Furthermore, as we continue to fine-tune for a longer number of steps, the quality of images will not reduce because we unlearn only along a precomputed direction (from Equation 3).

After unlearning for 800 steps, the FID (lower scores represent higher fidelity) using our method was 6.98 as opposed to 49.1 from the baseline method. The FID was computed using 10000 samples for each of the unlearned models. However, lower learning rates using the baseline method can avoid distortion but achieve little to none unlearning as seen in Figure 13.

C.6 PROMPT ENGINEERING DURING EVALUATION

During unlearning evaluation, the "surprised" feature was evaluated with the text caption "surprised with mouth open" since the surprised edit using the latent mapper generates images of faces with their mouth open. Unlike CLIP's text encoder, the VQA models can capture the image-text alignment better with a more detailed prompt. We suggest using this approach when evaluating other fine-grained edits as the objective is not to evaluate the VQA model, but to evaluate the image-text alignment before and after unlearning. All other prompts in the paper were evaluated with the same captions used for unlearning (*e.g.*, "purple hair", *etc.*)

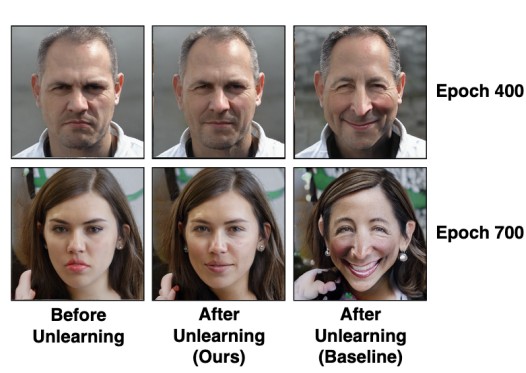

Figure 16: Qualitative comparison between directional unlearning (ours) and baseline method for the prompt "angry". Left most image was generated using a latent mapper trained on "angry".

