# OpenReview forum: "Text-to-Unlearn: Robust Concept Removal in GANs via Text Prompts"
_ICLR.cc/2026/Conference — ICLR 2026 Conference Withdrawn Submission_

### Official Review · Reviewer_vZR8 · 2025-10-29

**Soundness:** 3
**Presentation:** 3
**Contribution:** 3
**Rating:** 4
**Confidence:** 4

**Summary:**

The paper proposes Text-to-Unlearn, a method to remove specific concepts from pre-trained GANs using only text prompts. It relies on StyleCLIP to first train a latent mapper in the W+ space of the GAN that can edit images along the target concept to remove. The original and edited images are then used to find the direction associated with the target concept in the CLIP embedding space. The mapping network of the GAN is subsequently fine-tuned such that the direction between edited images generated via the fine-tuned GAN and the original GAN is aligned with the concept direction. This enforces the images generated via the fine-tuned model to be without the concept. The paper proposes different metrics, like the degree of unlearning and robustness analysis, to measure the effectiveness of the proposed method.

**Strengths:**

1. The proposed method extends unlearning beyond diffusion models, is innovative, and performs better than the more obvious baseline of maximizing the CLIP distance between the target concept prompt and generated images.
2. The paper also proposes metrics like the degree of unlearning and robustness analysis for a comprehensive evaluation.

**Weaknesses:**

1. The approach relies heavily on StyleCLIP for both discovering the concept direction and verifying the unlearning. While the results suggest that the model can no longer reproduce the target concept using StyleCLIP, it is unclear whether the underlying concept feature is fully erased. Extending the robustness analysis to include adversarial optimization directly on the fine-tuned model (e.g., optimizing a ΔW direction for individual image latents to generate the target concept by measuring the generated image similarity to the target prompt in some VLM embedding space) could strengthen the claims. Evaluating adversarial attacks using alternative vision-language models (e.g., SigLIP) would further validate the robustness of unlearning beyond CLIP space.
2. The LPIPS ablation (Appendix Fig. 15) is somewhat under-explained. Since LPIPS likely stabilizes training and maintains image realism, additional qualitative comparisons and a short discussion on its interaction with the directional loss and why it reduces the unlearning efficacy would make the analysis more insightful.
3. The current evaluation is restricted to StyleGAN2 on FFHQ, focusing on facial attributes, emotions, and identities. Demonstrating generalization on non-face datasets such as LSUN or AFHQ would provide stronger evidence that the approach extends beyond human faces.
4. The baseline comparison that maximizes CLIP distance (Eq. 11) does not include perceptual regularization terms such as LPIPS loss. Adding this to the baseline would provide a fairer comparison with the proposed directional loss.

**Questions:**

Please look at the weakness section, particularly regarding the reliance on StyleCLIP and CLIP for unlearning and evaluation (including robustness analysis).

---

### Official Review · Reviewer_G8pU · 2025-11-04

**Soundness:** 3
**Presentation:** 3
**Contribution:** 2
**Rating:** 4
**Confidence:** 2

**Summary:**

This paper proposes Text-to-Unlearn, a text-only framework for selectively erasing concepts from pre-trained GANs. It trains a StyleCLIP latent mapper for a target prompt, uses the mapper to create “with/without feature” image pairs, computes a reference edit vector in CLIP image-embedding space, and then fine-tunes a copy of the generator so that applying the same edit no longer moves along that vector—while regularizers preserve content. The method evaluates unlearning via vision-language scores and stress-tests robustness with a prompt-boundary attack that searches paraphrases near the original prompt. Results show targeted concept removal with limited collateral drift on face models.

**Strengths:**

1. The paper presents a text-only GAN unlearning pipeline that leverages a CLIP-space reference direction; the mechanism is well-motivated and straightforward to follow.
2. Robustness is assessed via a prompt-boundary attack (an adversarial prompt search around the target prompt) and the results are promising.

**Weaknesses:**

1. By design, the method’s effectiveness hinges on CLIP’s text/image embeddings, both for the directional loss and for the prompt neighborhood in the boundary attack. Unlearning quality can suffer when the reference direction is not sufficiently input-agnostic or when a concept is poorly represented (as acknowledged in Section 7).
2. StyleCLIP’s latent mapper is trained once per prompt; scaling to many concepts/paraphrases or enabling dynamic unlearning can be costly.
3. As mentioned in 1., the prompt-boundary attack considers only CLIP-nearest paraphrases, whose semantic fidelity is not guaranteed, and the examples cover only a little of the prompt space for a given concept. More verbally/semantically diverse prompts, varying in description, length, and style, would better demonstrate robustness.

**Questions:**

1. Could the authors report the computational cost and wall-clock time of the proposed method, and compare these against the baselines?
2. Could the authors comment on how the proposed method is inherently tied to GANs, or can it be extended to other generative frameworks?

---

### Official Review · Reviewer_YCYB · 2025-11-05

**Soundness:** 3
**Presentation:** 3
**Contribution:** 2
**Rating:** 2
**Confidence:** 4

**Summary:**

This paper proposes a method to unlearn concept for GANs by finetuning the generation model. It combines the semantic direction provided by StyleCLIP from text prompt and the few-shot adaptation scheme from StyleGAN-NADA. The losses for unlearning attributes and id from the generated face images are proposed. Experiments are conducted on CelebA-HQ to validate the effectiveness of the proposed method with qualitative results and self-proposed metric of degree of unlearning.

**Strengths:**

- The topic of unlearning concept on GANs is less studied compared to diffusion based models.
- The proposed method of using text prompts to unlearn concept sounds technically feasible.

**Weaknesses:**

- Both of the techniques supporting the proposed method come from existing literature. This makes the technical contribution of this paper very marginal if there is a technical contribution.
- The challenges listed in Sec. 3.2 are not fully addressed by the proposed method. To be specific, the first challenge was handled by the latent space itself rather than the proposed method in this paper. The second challenge is addressed by the adopted work of StyleGAN. The third challenge is addressed by the proposed metric, but not satisfy me (see the next comment).
- The proposed metric to measure the extent to which unlearning is successful is not very straightforward and sensible to me by using the Wasserstein-1 distance. I would suggest a metric related to the classifier based metric, which could classify if the image contains that concept or not, which is more robust and sensible.
- Fig. 1 should be self-contained (by adding more explanations).

**Questions:**

- The task of concept removal is well received. However, the reason to use text prompt should be well justified.
- See weaknesses above.

---

### Official Review · Reviewer_RHsp · 2025-11-05

**Soundness:** 2
**Presentation:** 3
**Contribution:** 2
**Rating:** 4
**Confidence:** 3

**Summary:**

This paper introduces Text-to-Unlearn, a two-phase, text-guided framework that removes concepts from a pre-trained StyleGAN without the use of any curated datasets. The method first computes a CLIP edit direction for the target concept using a latent mapper and a frozen copy of the generator, then fine-tunes the trainable generator so its outputs move against that direction, while LPIPS and ID losses preserve appearance and utility. The approach handles features (e.g., hairstyles, accessories), identities (e.g., celebrities), and expressions, including multi-attribute prompts. The paper also proposes a quantitative metric ie degree of unlearning, built from VLM alignment distributions, and a prompt boundary attack that perturbs the text embedding to probe the robustness of the unlearned concept.

**Strengths:**

- The paper proposes a simple, cross-modal unlearning that needs only text for GAN architecture based generative models. The CLIP-directional fine-tuning is conceptually clean and avoids dataset collection.
- The paper stress tests the proposed approach through the adversarial robustness. The prompt boundary attack perturbs token embeddings within an neighborhood and evaluates recovery of erased concepts; the paper also defines a robustness score.
- The white-box threat model is clearly mentioned.
- The pseudo code for the algorithm is written in the paper.

**Weaknesses:**

- Some ablation on the choice of text embedding like CLIP dependence and bias would be great to have for the readers. Because both the direction and evaluation rely on CLIP/VLMs, coverage gaps or biases can limit unlearning, would be great to have some discussions around this.
- The empirical results are confined to StyleGAN2 on limited datasets like FFHQ (faces) , CelebA-HQ; it’s unclear how well the method extends to non-face domains, also other GANs, or diffusion/flow generators that dominate modern pipelines.
- The paper proposes a per-prompt mapper training that introduces a overhead. The pipeline trains a latent mapper for each prompt with tuned hyperparameters, which adds operator effort and may affect scalability.
- It would be great to see more evaluations to validate the efficacy of the approach, A small human study or downstream task check (post-unlearning editability, identity verification thresholds) could calibrate how γ and
- The related work sections feels a bit limited and would be great to have more modern concept removal references for diffusion models too (like Ablating Concepts in Text-to-Image Diffusion Models, Nupur et al, ICCV 2023)

**Questions:**

- Would GAN inversion techniques be possible for a given image containing the target concept that was removed, for the fine-tuned target GAN

---

### Note · Authors · 2025-11-12

I have read and agree with the venue's withdrawal policy on behalf of myself and my co-authors.